# On the Possible Asymmetry in Gamma Rays from Andromeda Due to Inverse Compton Scattering of Star Light on Electrons from Dark Matter Annihilation

**Konstantin Belotsky [1,2,*] and Maxim Solovyov [1]**

[1] Department of Elementary Particle Physics, National Research Nuclear University MEPhI (Moscow Engineering Physics Institute), Kashirskoe Shosse 31, 115409 Moscow, Russia; max07s@mail.ru

[2] Laboratory of Cosmology and Particle Physics, Novosibirsk State University, Pirogova Street 1, 630090 Novosibirsk, Russia

[*] Correspondence: k-belotsky@yandex.ru

**Abstract:** Dark matter is a popular candidate to a new source of primary-charged particles, especially positrons in cosmic rays, which are proposed to account for observable anomalies. While this hypothesis of decaying or annihilating DM is mostly applied for our Galaxy, it could possibly lead to some interesting phenomena when applied for the other ones. In this work, we look into the hypothetical asymmetry in gamma radiation from the upper and lower hemisphere of the dark matter halo of the Andromeda galaxy due to inverse Compton scattering of starlight on the DM-produced electrons and positrons. While our 2D toy model raises expectations for the possible effect, a more complex approach gives negligible effect for the dark halo case, but shows some prospects for a dark disk model.

**Keywords:** cosmic gamma-radiation; Andromeda galaxy; dark matter

## 1. Introduction

Decaying or annihilating dark matter particles are often involved in accounting for unusual observational effects in our Galaxy, such as the positron excess in cosmic rays. However, it would be baseless to expect the considered DM properties to be specific only to the Milky Way. And. while there are constraints upon the DM cross-section of the annihilation and decay mean lifetime placed by observations of gamma-rays from dwarf galaxies, it is possible that such processes might lead to other peculiar observable effects outside of our Galaxy.

In this regard, M31 (Andromeda galaxy) is quite promising. It is one of the few galaxies that can be observed in gamma radiation via modern and upcoming experiments such as satellite experiments like Fermi-LAT [1,2] and Gamma-400 [3], and ground-based ones like HAWC [4], HESS [5], MAGIC [6], LHAASO [7], and VERITAS [8]. The observations already show the signs of possible gamma-ray excess from the Andromeda halo that could possibly be connected to DM [9]. And, while the current experiments often lack the angular resolution, some of the forthcoming ones, such as satellite gamma-ray telescope project Gamma-400 [3], would have it covered, allowing for an in-depth study of the spatial distribution of gamma radiation from M31.

In this work, we explore one possible effect connected with gamma-radiation spatial distribution. If there is a source (dark matter annihilation) of high-energy-charged particles (positrons and electrons) in the Andromeda halo, then the gamma rays produced by inverse Compton scattering of starlight on them could show some anisotropy due to the 15° inclination of the galaxy disk with respect to the observer's line of sight leading to slightly different predominant scattering angles for the «upper» and «lower» hemispheres of the Andromeda halo. Here, we will briefly recap and extend the results obtained for

our simplistic 2D toy model of such process [10], that served as a motivation for this work, and then proceed to study the case in the frame of a much more complex 3D model. This consideration differs from [11] where the effect has been estimated with the help of existing programs of numerical calculation of CR propagation in our Galaxy, while here the effect is considered explicitly at a simplified physical level.

## 2. 2D Toy Model

In our model, initial photons of the starlight are emitted from the center of M31, represented by a thick black line in Figure 1, perpendicular to the galactic disk. They are subjected to inverse Compton scattering on the electrons or positrons at the equidistant points $A_+$ and $A_-$ in the upper and lower hemispheres of the halo, respectively. The observer is considered to be infinitely far from M31, and therefore, the scattered photons are parallel, making the scattering angle to be 75° for the upper and 105° for the lower one. In this work, we set the initial photon energy to be 1–2 eV, while electron energy is varied.

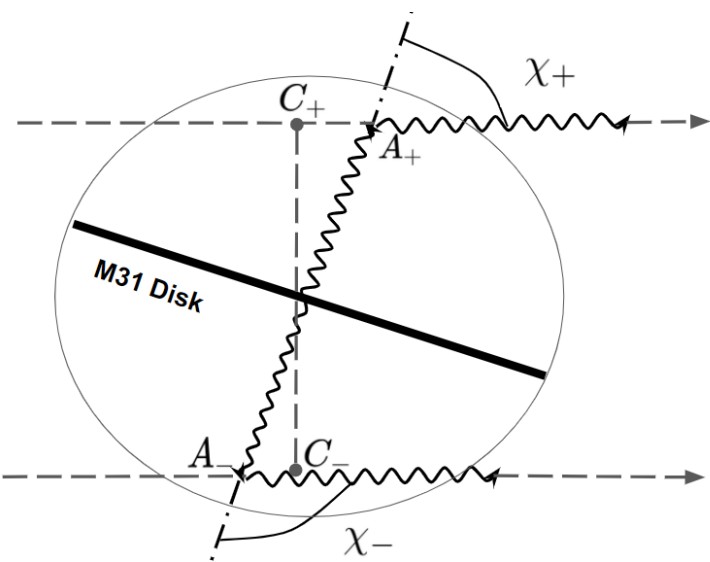

**Figure 1.** The scheme for scattering process in upper and lower Andromeda hemispheres with the chosen points. Observer is on the right.

To calculate the parameters of final photons, we use a well-known Compton scattering formula along with the Lorentz boosts and Lorentz invariance of scalars:

$$
\begin{cases}
E_\gamma^{\text{lab}} = \dfrac{E_{\gamma 0}^{\text{lab}}}{1 + \dfrac{E_{\gamma 0}^{\text{lab}}}{m_e}(1 - \cos(\theta_{\text{lab}}))} \\
E_\gamma^{\text{lab}} = \gamma(E_\gamma - \vec{p}_\gamma \cdot \vec{v}_{e0}) \\
E_{\gamma 0}^{\text{lab}} = \gamma(E_{\gamma 0} - \vec{p}_{\gamma 0} \cdot \vec{v}_{e0}) \\
\vec{p}_{\gamma 0}^{\,1} = \vec{p}_{\gamma 0} + (\gamma - 1)(\vec{p}_{\gamma 0} \cdot \vec{n})\vec{n} - \gamma E_{\gamma 0}\vec{v}_{e0} \\
\vec{p}_\gamma^{\,1} = \vec{p}_\gamma + (\gamma - 1)(\vec{p}_\gamma \cdot \vec{n})\vec{n} - \gamma E_\gamma \vec{v}_{e0} \\
\vec{p}_{\gamma 0} \cdot \vec{p}_\gamma = \vec{p}_{\gamma 0}^{\,1} \cdot \vec{p}_{\gamma 0}^{\,1}.
\end{cases}
\tag{1}
$$

Here, $E_{\gamma 0}$ and $E_\gamma$ are, respectively, the initial and final energies of the photon in the observer frame of references; $E_{\gamma 0}^{lab}$ and $E_\gamma^{lab}$ are those in the laboratory frame of reference, where the electron is in the rest; $m_e$ is the electron mass; $\theta_{lab}$ is the angle of scattering in the laboratory frame of reference; $\gamma$ is the gamma factor of initial electron; $\vec{v}_{e0}$ is its velocity; and $\vec{p}_{\gamma 0}$, $\vec{p}_\gamma$, $\vec{p}_{\gamma 0}^{\,l}$, and $\vec{p}_\gamma^{\,l}$ are, respectively, the momenta of the initial and scattered photon in the observer and laboratory frame of reference, and $\vec{n} = \dfrac{\vec{v}_{e0}}{|\vec{v}_{e0}|}$.

By solving the system above, it is possible to obtain the equation for the energy of the final photon in dependence on the initial parameters of the involved particles. However, here we would focus on the case of the maximum possible energy of the final photon that corresponds to the case of the initial electron traveling in the same direction as the final photon, e.g., directly towards the observer (horizontally to the right in Figure 1. Under this assumption, it will be the following:

$$E_\gamma = \frac{E_{\gamma 0}(1 - v_{e0} \cos \theta)}{1 - v_{e0} + \frac{E_{\gamma 0}}{\gamma m_e}(1 - \cos \theta)} = \frac{(1 + v_{e0})\gamma^2 E_{\gamma 0}(1 - v_{e0} \cos \theta)}{1 + (1 + v_{e0})\gamma \frac{E_{\gamma 0}}{m_e}(1 - \cos \theta)}, \tag{2}$$

where $\theta$ is the scattering angle in the observer's reference frame that equals $\chi_+ = 75°$ for scattering in the upper hemisphere of the M31 halo and $\chi_- = 115°$ for the lower one. The second form is obtained by multiplying both parts of the fraction by $\gamma^2$ for easier dealing with the case of the relativistic initial electron. To estimate the anisotropy for the value of maximum photon energy, we use

$$R = \frac{E_\gamma^+}{E_\gamma^-}, \tag{3}$$

where $E_\gamma^+$ and $E_\gamma^-$ correspond to the upper and lower hemispheres, respectively.

Figure 2 shows the dependence of this ratio $R$ from the gamma factor of the initial electron. At $\gamma = 1$, corresponding to $v_e = 0$, Equation (2) takes the form

$$E_\gamma = \frac{E_{\gamma 0}}{1 + \frac{E_{\gamma 0}}{m_e}(1 - \cos \theta)} \approx E_{\gamma 0}$$

since $E_{\gamma 0} \ll m_e$, therefore, $R = 1$. At $1 \ll \gamma \ll \frac{m_e}{E_{\gamma 0}}$, corresponding to $m_e \ll E_e \ll 250$ GeV for the initial photon energy of 1 eV, $v_e \approx 1$, the denominator in Equation (2) can be approximated to a unity; therefore, $E_\gamma \approx 1 - \cos \theta$ and $R = \frac{1 - \cos \chi_+}{1 - \cos \chi_-} \approx 0.6$. Lastly, when $\gamma \gg \frac{m_e}{E_{\gamma 0}}$,

$$E_\gamma \approx \frac{2\gamma^2 E_{\gamma 0}(1 - \cos \theta)}{2\gamma \frac{E_{\gamma 0}}{m_e}(1 - \cos \theta)} = \gamma m_e,$$

and $R = 1$ again.

The anisotropy of gamma-ray fluxes at the highest energy can be roughly estimated as

$$R_\Phi = \frac{\frac{d\sigma}{d\Omega}A_+}{\frac{d\sigma}{d\Omega}A_-}, \tag{4}$$

where $\frac{d\sigma}{d\Omega}$ is the cross section of the process in the observer reference frame that can be calculated as

$$\frac{d\sigma}{d\Omega} = \frac{d\sigma}{d\Omega^l}\left|\frac{d\Omega^l}{d\Omega}\right|, \tag{5}$$

$$\frac{d\sigma}{d\Omega^l} = \frac{1}{2}r_e^2 \lambda^2 \left(\lambda + \frac{1}{\lambda} - 1 + \cos^2 \theta_{lab}\right), \tag{6}$$

$$\frac{d\Omega^l}{d\Omega} = \frac{d\theta_{lab}}{d\theta} = \frac{d\theta_{lab}}{d\cos \theta_{lab}}\frac{d\cos \theta_{lab}}{d\cos \theta}\frac{d\cos \theta}{d\theta} = \frac{\sin \theta}{\sin \theta_{lab}}\frac{d\cos \theta_{lab}}{d\cos \theta}, \tag{7}$$

where $\frac{d\sigma}{d\Omega^l}$ is the cross section of the process in the laboratory reference frame defined by the Klein–Nishina Formula (6), with $r_e$ being the classical electron radius and $\lambda = \frac{E_\gamma^l}{E_{\gamma 0}^l}$,

while $\frac{d\Omega^l}{d\Omega}$ defines its changes due to the transition into the observer's frame of reference. The values of $\sin\theta_{lab}$ and the functional dependence between $\cos\theta_{lab}$ and $\cos\theta$ can be obtained from (1).

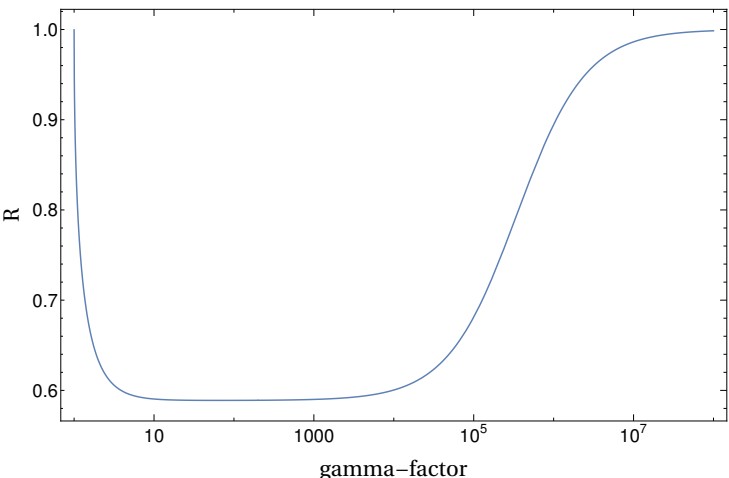

**Figure 2.** Dependence of $R$ of Equation (3) showing asymmetry effect for maximal photon energy between upper and lower hemispheres of Andromeda galaxy from $\gamma$-factor of initial electron.

Figure 3 demonstrates the obtained results. However, unlike the ratio for the values of maximal energies, this one was obtained using a numerical calculation approach that starts giving unstable results for high values of the gamma factor. Due to that, its upper limit was set to be $10^3$ for this case.

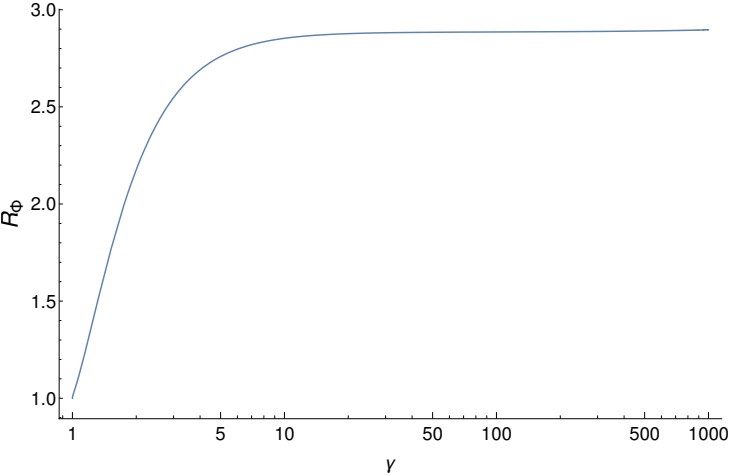

**Figure 3.** Dependence of $R_\Phi$ showing asymmetry effect for flux of gamma radiation with maximal photon energy between upper and lower hemispheres of Andromeda galaxy from $\gamma$-factor of initial electron.

## 3. 3D Model

### 3.1. Halo Case

Suggested 2D model produces very interesting and promising results that, however, need to be treated cautiously due to its simplistic nature. In reality, the whole disc of the Andromeda galaxy serves as the source of photons which are emitted in all directions and can be scattered anywhere inside the halo. And, the observer with a fixed line of sight can simultaneously detect the photons originated from different parts of the galactic disc and scattered in different areas along their line of sight. Moreover, our main interest lies in expanding to the case of M31 of the models used to describe the excess of charged particles

in our Galaxy [12–14], and such models need electrons and positrons with energies up to 2 TeV, where the anisotropy is expected to vanish.

To address these issues, we considered a more complex 3D model that mainly focuses on the electrons with an energy of 1.8 TeV (corresponding to the direct annihilation of DM particles with the same mass considered in our works concerning the DAMPE experiment [13,14]) and briefly touches the ones in the MeV energy range. Andromeda is modeled as a homogeneous thin disk with a radius $R_A = 30$ kpc, located at a distance $d = 800$ kpc from the observer with an inclination of 15°, with respect to their line of sight towards its center. It emits photons with an energy of 2 eV, each isotropically in every direction, with a total luminosity $L_A = 2.6 \times 10^{10} L_\odot$, where $L_\odot = 3.8 \times 10^{26}$ W is the solar luminosity. It is surrounded by a dark matter halo with $R_{DM} \sim 100$ kpc with an NFW [15] density profile

$$\rho_{\text{NFW}}(r) = \frac{\rho_0}{\frac{r}{R_s}\left(1 + \frac{r}{R_s}\right)^2},$$ (8)

where $\rho_0 = 0.25$ GeV·cm$^{-3}$ and $R_s = 24$ kpc, which, for our Galaxy, corresponds to a DM density of 0.4 GeV·cm$^{-3}$ near the Sun. For the calculation, we suppose that dark matter consists of particles with a mass value (1) $M_X = 1.8$ TeV or (2) 100 MeV that can annihilate directly into an electron–positron pair with a cross section $\langle \sigma v \rangle = 10^{-23}$ cm$^3$·s$^{-1}$, of which the value is taken from our models for our Galaxy. Here, we neglect the process of final-state radiation and the traveling path of electrons and positrons before the scattering is considered negligible; therefore, their energy equals to $M_X$. They are considered to have an isotropic distribution of their momenta directions. The direction of the observer's line of sight is described by two angles—longitude $l$ and latitude $b$.

Like in the case of the 2D model, here we focus on the asymmetry in the values of photon maximal energy and its corresponding fluxes. For the former, the main Equation (2) is still applicable, as it was obtained without respect to the model's geometry, only requiring the initial electron to have the same direction as the final photon. However, here we focus on ultra-relativistic initial electrons, so that the formula can be simplified with expanding $v_{e0}$ into series:

$$v_{e0} = \frac{p_{e0}}{E_{e0}} = \frac{\sqrt{E_{e0}^2 - m_e^2}}{E_{e0}} \approx \left(1 - \frac{m_e^2}{2E_{e0}^2}\right),$$

resulting in

$$E_\gamma \approx \frac{2E_{e0}^2 E_{\gamma 0}\left(1 - \cos\theta + \frac{m_e^2}{2E_{e0}^2}\cos\theta\right)}{m_e^2 + 2E_{\gamma 0}E_{e0}(1 - \cos\theta)}.$$

However, the third term in the numerator is relevant when $\cos\theta \to 1$, which corresponds to the $E_\gamma \to E_{\gamma 0}$ that lies out of our sphere of interest, so it can be omitted. This gives the following formula for the maximal photon energy:

$$E_\gamma \approx \frac{2E_{e0}^2 E_{\gamma 0}(1 - \cos\theta)}{m_e^2 + 2E_{\gamma 0}E_{e0}(1 - \cos\theta)} = \frac{E_{e0}(1 - \cos\theta)}{\frac{m_e^2}{2E_{\gamma 0}E_{e0}} + (1 - \cos\theta)}.$$ (9)

In this 3D model, however, $\theta$ is not considered to be constant and can be varied, depending on the coordinates of the point of initial photon production and the point of scattering. The value of the energy rises with the decrease in $\cos\theta$, so the maximization of $E_\gamma$ from Equation (9) is reduced to maximizing the scattering angle (which can change in range from zero to 180°) for each observer's line of sight with specific angular coordinates $b$ and $l$. It was found that $\theta$ reaches the maximum if scattering takes place in the furthest possible point along the line of sight, while the photon starting point is situated at the disk rim on its lower part with a slight shift from the lowest point to the opposite direction in respect to the line of sight (see Figure 4). However, it was found that this shift from

the lowest point of Andromeda changes the energy value insignificantly, and therefore, to simplify the calculations, the latter was used as the approximation.

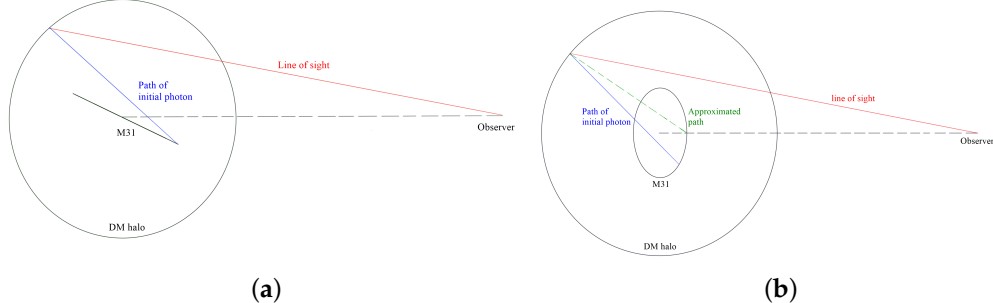

**(a)**             **(b)**

**Figure 4.** Scattering scheme, viewed from the side (**a**) and the top (**b**).

The obtained results are shown in Figure 5 in the form of contour plots of the energy value dependent on the longitude and latitude for both values of electron energy (i.e., DM particle mass). The values of $R$ dependent on latitude $b$ for $l = 0$ are shown in Figure 6.

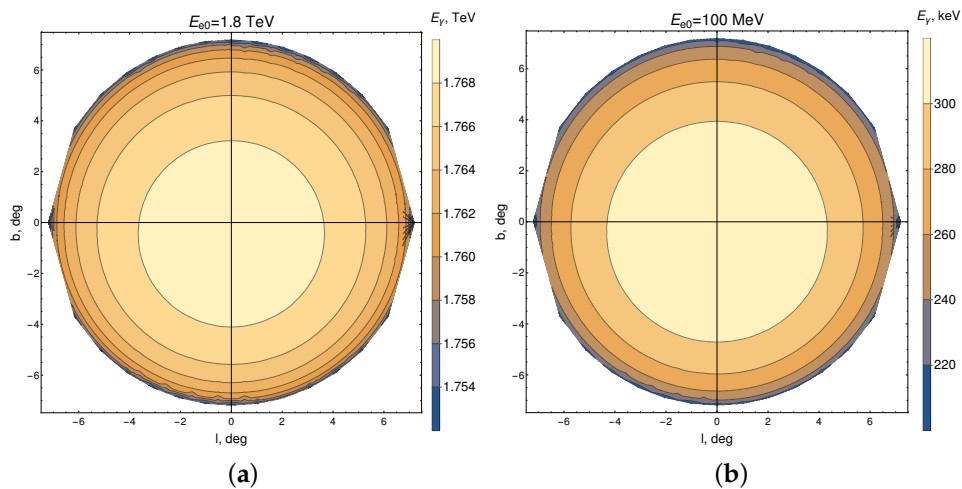

**(a)**             **(b)**

**Figure 5.** Value of maximal photon energy dependent on line of sight directions for initial electron with energy of 1.8 TeV (**a**) and 100 MeV (**b**).

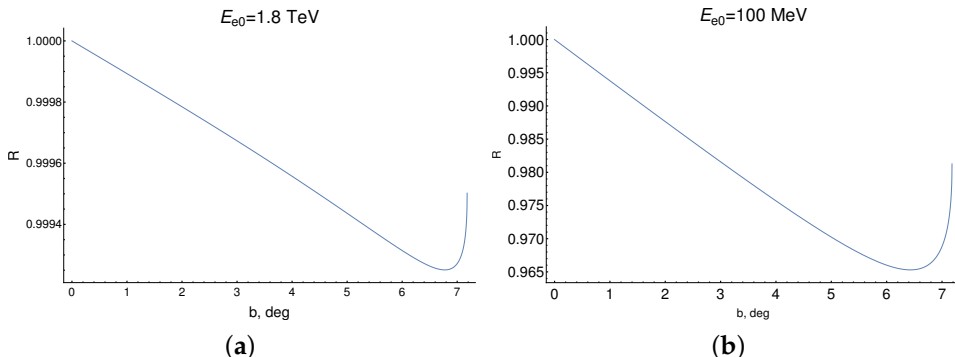

**(a)**             **(b)**

**Figure 6.** Ratio $R$ for maximal energy from upper hemisphere to the one from lower hemisphere dependent on latitude $b$ for $l = 0$ and initial electron with energy of 1.8 TeV (**a**) and 100 MeV (**b**).

As can be seen from the figures, the case of $E_e = 1.8$ TeV leads to very high energy of gamma radiation and, unsurprisingly, very low asymmetry. In the MeV range, the final photon holds a much smaller part of the initial electron energy and the difference between the hemispheres reaches 3%, unlike the 40% in the 2D model.

For the flux asymmetry, the much more complex approach compared to the 2D model was adopted.

$$R_\Phi = \frac{\Phi_+}{\Phi_-} \tag{10}$$

For a more general case with an unrestrained initial electron direction, the expression for the flux was constructed as follows:

$$\Phi(E_\gamma) = \frac{dN}{dE \cdot dt \cdot dS \cdot d\Omega} =$$

$$= \frac{1}{\Delta\Omega_{\text{l.o.s.}}} \int_{\text{l.o.s.}} r^2 dr d\Omega_{\text{l.o.s.}} \int_{\text{Andr}} dS_{\text{Andr}} \int_{\text{el. dir}} d\Omega_{\text{e}} \frac{dn_e}{d\Omega_{\text{e}}} \times \tag{11}$$

$$\times \frac{dn_\gamma}{dS_{\text{Andr}}} \cdot \frac{d\sigma}{d\Omega_{\text{scattering}}} \cdot \Delta\Omega_{\text{scattering}} \cdot v_{\text{rel}} \cdot \frac{1}{\Delta\Omega_{\text{scattering}} r^2} \cdot \frac{dN}{dE}$$

It is basically a heavily modified version of the expression used to calculate primary gamma radiation from DM annihilation in the case of our Galaxy, see Equation (2) from [13] for example. It divides the sky around M31 into «cells» or bins with an angular size of $\Delta\Omega_{\text{l.o.s.}}$ corresponding to the line of sight directed at its center. The first integration corresponds to the scattering point and goes over such bins and the distance from the observer $r$ is confined by the DM halo edges. The second integration corresponds to the starting point of the initial photon and is taken over by the whole M31 disk. The third one goes over all possible directions of the initial electron. The term $\frac{dn_e}{d\Omega_e}$ represents the number density of the electrons with directions inside specific $d\Omega_e$; $\frac{dn_\gamma}{dS_{\text{Andr}}}$ is the fraction of the total number density of photons produced by a specific portion of the M31 disk $dS_{\text{Andr}}$; $\frac{d\sigma}{d\Omega_{\text{scattering}}} \cdot \Delta\Omega_{\text{scattering}}$ is the cross-section for the specified initial particles to produce a final photon with a direction lying inside the $\Delta\Omega_{\text{scattering}}$; $v_{\text{rel}} = c$ – is the relative speed of the particles; $\Delta\Omega_{\text{scattering}} r^2$ is the area that the final photons end on near the observer; and $\frac{dN}{dE}$ is their spectrum.

The number density of electrons (and positrons) can be obtained using their production rate:

$$\frac{dn_e}{d\Omega_e} = \frac{d^2 n_e}{d\Omega_e dt}\Delta t = \frac{1}{4\pi} 2\frac{\rho_{DM}^2}{4M_X^2}\langle\sigma v\rangle_{DM} \times \frac{l_e}{c} = \frac{\rho_{DM}^2}{8M_X^2}\langle\sigma v\rangle_{DM} \times \frac{1}{cn_\gamma\sigma}, \tag{12}$$

where factor 2 takes into account that the $e^+e^-$ pair is produced in one reaction, $l_e$ is the electron-free path, and $n_\gamma$ and $\sigma$ are the total number density of photons and the total scattering cross-section, respectively. The latter can be given by

$$\sigma = \begin{cases} \pi r_e^2 \frac{m_e}{E_{\gamma 0}^{\text{lab}}}\left(\log\left(2\frac{E_{\gamma 0}^{\text{lab}}}{m_e}\right) + \frac{1}{2}\right), E_{\gamma 0}^{\text{lab}} \gg m_e \ (M_X = 1.8 \text{ TeV}) \\ \frac{8}{3}\pi r_e^2\left(1 - 2\frac{E_{\gamma 0}^{\text{lab}}}{m_e}\right), E_{\gamma 0}^{\text{lab}} \ll m_e \ (M_X = 100 \text{ MeV}). \end{cases} \tag{13}$$

While for the both energy ranges the initial electrons are ultrarelativistic, the transition to the laboratory frame of reference leads to strongly different values of $E_{\gamma 0}^{\text{lab}}$, therefore requiring different approximations of the used cross section as shown in the expression above.

The number density of the photons can be obtained in the following:

$$n_\gamma(\vec{r}) = \int_0^{R_A} \int_0^{2\pi} r_a \frac{dn_\gamma}{dS_{\text{Andr}}} dr_a d\varphi_a, \tag{14}$$

where $r_a$ and $\varphi_a$ are coordinates of the initial photon emission point in the polar coordinate system associated with the Andromeda disk. Then, the photon number density fraction produced by $dS_{\text{Andr}}$ around it at the scattering point with coordinates $\vec{r}$ is given by

$$\frac{dn_\gamma}{dS_{\text{Andr}}}(\vec{r}) = \frac{L_A}{4\pi R_A^2 E_{\gamma 0}} \cdot \frac{1}{4\pi c |\vec{r}_a - \vec{r}|^2} \tag{15}$$

The differential cross section is calculated similarly to Equation (5):

$$\frac{d\sigma}{d\Omega_{\text{scattering}}} = \frac{d\sigma}{d\Omega_{\text{lab}}} \cdot \left| \frac{d\Omega_{\text{lab}}}{d\Omega_{\text{scattering}}} \right| \tag{16}$$

with

$$\left| \frac{d\Omega_{\text{lab}}}{d\Omega_{\text{scattering}}} \right| = \left| \frac{\partial \cos(\theta_{\text{lab}})}{\partial \cos(\theta)} \right| \cdot \left| \frac{\partial \varphi_{\text{lab}}}{\partial \varphi} \right|. \tag{17}$$

The final photon spectrum is given by

$$\frac{dN}{dE} = \delta(E - E_\gamma), \tag{18}$$

where $E_\gamma$ is the function of coordinates of the photon emission point and the scattering point obtained from solving the system (1) in the general case and the calculation of the scattering angle from the given coordinates.

In principle, Formula (11) allows us to calculate the fluxes for the whole spectrum of the final photons. However, it was found that there is a strong resonance-like dependence of the final photon energy on the co-alignment of its momentum to the momentum of the initial electron. Paired with seven-dimension integration, it makes the proper calculation very demanding considering the computation power and time. Due to that, we focused on the gamma flux corresponding to the highest energies, like in the case of the 2D model.

To simplify the calculation, we consider the integration term to be constant for the range of integration parameters corresponding to the energy range of $E_\gamma^{\text{max}} - \Delta E$ to $E_\gamma^{\text{max}}$, allowing us to substitute the integrations with multiplication of the said term by the parameter ranges $\Delta r$ and $\Delta \Omega_e$, while for the simplicity case, the whole Andromeda disk was considered to be the source of initial photons. For this work, $\Delta E$ was chosen to be equal $0.01 E_\gamma^{\text{max}}$. The actual calculation was performed numerically using the grid that divides each parameter range (line-of-sight coordinate $r$; initial photon birthplace coordinates $r_A$ and $\varphi_A$; and angles $\theta_e$ and $\varphi_e$ defining the initial electron direction) into 10–12 steps:

$$\Phi \approx \frac{c}{\Delta E_\gamma} \sum_{i=1}^{10} \sum_{j=1}^{10} \sum_{k=1}^{12} \sum_{n=1}^{12} \sum_{m=1}^{10} \Delta r^i \Delta \Omega_e^{nm} \Delta S_{Andr}^{jk} \times$$
$$\times \frac{dn_e}{d\Omega}(r^i, \varphi_e^n, \theta_e^m) \frac{dn_\gamma}{dS_{\text{Andr}}}(r^i, r_A^j, \varphi_A^k) \frac{d\sigma}{d\Omega_s}(r^i, r_A^j, \varphi_A^k, \varphi_e^n, \theta_e^m), \tag{19}$$

However, even with such simplifications, this method remains very hungry for computation power and is time-consuming; therefore, it is not very suitable for detailed analysis.

Obtained estimations are given in Table 1. As could be seen, for the halo case, the anisotropy emerges only for the low mass of initial particle that lies outside of our main interest.

**Table 1.** Maximum final photon energy and corresponding fluxes in the halo case.

| $M_X$ | $E_+$ | $E_-$ | $E_+^2\,\Phi_+$ | $E_-^2\,\Phi_-$ | $R$ | $R_\Phi$ |
|---|---|---|---|---|---|---|
| 1.8 TeV | 1777 GeV | 1777 GeV | $5\times10^{-11}\,\frac{\text{GeV}}{\text{cm}^2\,\text{s sr}}$ | $5\times10^{-11}\,\frac{\text{GeV}}{\text{cm}^2\,\text{s sr}}$ | 1 | 1 |
| 100 MeV | 270 keV | 280 keV | $4.5\times10^{-10}\,\frac{\text{keV}}{\text{cm}^2\,\text{s sr}}$ | $10^{-11}\,\frac{\text{keV}}{\text{cm}^2\,\text{s sr}}$ | 0.96 | 45 |

Disk Case

As it was shown above, in the case of TeV-scale energies of the initial electron, the anisotropy generated by the Andromeda stellar disk inclination is not enough to be observed. However, introducing a new source in the form of anisotropic dark matter spatial distribution might strengthen the effect. This part of the work is dedicated to testing this possibility.

The dark matter disk was approximated as an ellipsoid with 100, 100, and 10 kpc semi-axes (Figure 7) with a Read density profile [16]:

$$\rho(R,z) = \rho_0 e^{-\frac{R}{R_c}}\, e^{-\frac{|z|}{z_c}}, \tag{20}$$

with $z_c = 0.4$ kpc , $R_c = 7$ kpc, and $\rho_0 = 1.32$ GeV/cm$^3$. And, while the density profile and its parameters were directly taken from our works considering positron excess in our Galaxy, the total size of the disk was chosen from general considerations to make it distinctly anisotropic but not overly thin at the same time.

With this setup, from the observer's point of view, the angular size of the regions emitting gamma radiation are somewhat different for both hemispheres, with the upper being $\approx 1.78°$ and the lower being $2.22°$ (as can be seen from Figure 7b, where the upper and lower red lines correspond to the same angular size).

The final estimation for $b = \pm 1.75°$ are given in Table 2. A comparison to Table 1 shows that the introduction of an extra source of anisotropy in the form of disk-like DM spatial distribution allows some anisotropy, in the case of the heavy initial particle, and greatly increases the existing one, in the case of light one.

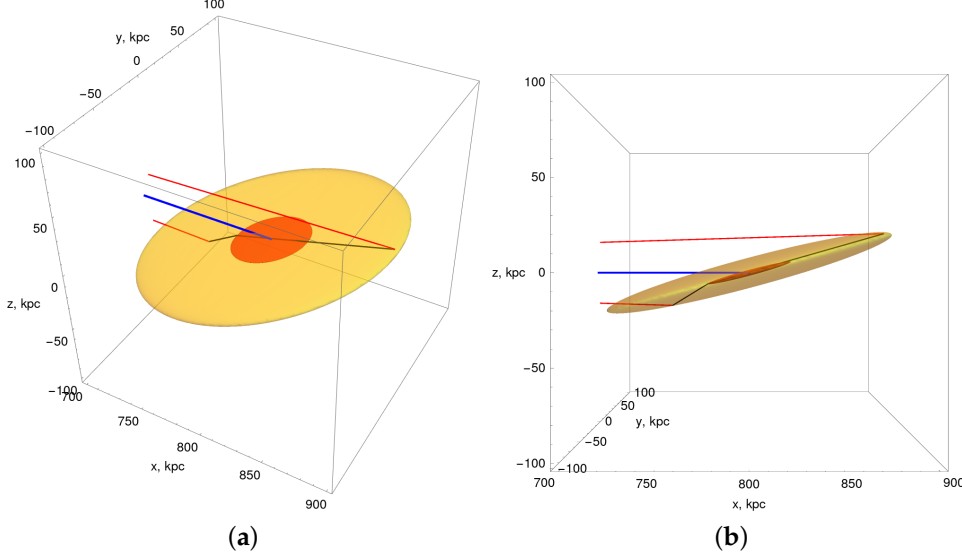

(**a**)　　　　　　(**b**)

**Figure 7.** Scattering scheme in the dark disk case, in perspective (**a**) and from the side (**b**). Dark matter disk is depicted as yellow, Andromeda stellar disk as orange, blue line connects the observer with Andromeda galactic center, red lines show the trajectory of the final photon with $b = \pm 1.75°$, black lines show the initial photon paths to the scattering points necessary to achieve maximum final energy.

**Table 2.** Maximum final photon energy and corresponding fluxes in the disk case.

| $M_X$ | $E_+$ | $E_-$ | $E_+^2\,\Phi_+$ | $E_-^2\,\Phi_-$ | $R$ | $R_\Phi$ |
|---|---|---|---|---|---|---|
| 1.8 TeV | 1777 GeV | 1578 GeV | $3.7 \times 10^{-28}\,\frac{\text{GeV}}{\text{cm}^2\,\text{s sr}}$ | $2.9 \times 10^{-28}\,\frac{\text{GeV}}{\text{cm}^2\,\text{s sr}}$ | 1.13 | 1.28 |
| 100 MeV | 315 keV | 39 keV | $2.3 \times 10^{-29}\,\frac{\text{keV}}{\text{cm}^2\,\text{s sr}}$ | $2.8 \times 10^{-38}\,\frac{\text{keV}}{\text{cm}^2\,\text{s sr}}$ | 8 | $10^9$ |

## 4. Conclusions

In this work, we considered the potential anisotropy in secondary gamma radiation in the case of applying the decaying or annihilating DM models developed to account for the charged particles' excess in CR from the Milky Way to the Andromeda galaxy. The effect was considered first using the simplistic 2D toy model, which gave the promising prediction for the asymmetry levels of the maximum photon energy and corresponding flux, promoting further research.

The follow-up study with the more accurate 3D model affirmed that in the halo case, some anisotropy lies in the range of the low mass of the DM particle and virtually vanishes at the TeV-scale masses we are interested in. However, the introduction of an additional source of anisotropy in the form of the dark matter disk significantly strengthened the effect and induced an anisotropy in the gamma-ray spatial distribution itself, at the cost of considerably lower predicted fluxes. Though, the technique used to obtain these estimations is not well suited to answer whether these effects could actually be detected in one form or another via modern or upcoming experiments; therefore, we require a separate study on that matter.

**Author Contributions:** K.B.—concept development, M.S.—implementation of calculation model. All authors have read and agreed to the published version of the manuscript

**Funding:** This work was supported by the Ministry of Science and Higher Education of the Russian Federation by project No. FSWU-2023-0068 "Fundamental and applied research of cosmic rays".

**Data Availability Statement:** Data sharing not applicable.

**Acknowledgments:** We would like to thank A. Egorov, M. Khlopov, M. Laletin, and S. Rubin for interest to the work and useful discussions and also D. Fargion for providing relevant references.

**Conflicts of Interest:** The authors declare no conflict of interest.

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
