# Peer review of "On the Possible Asymmetry in Gamma Rays from Andromeda Due to Inverse Compton Scattering of Star Light on Electrons from Dark Matter Annihilation"

_galaxies, doi:10.3390/galaxies11060109_

Round 1

Reviewer 1 Report

Comments and Suggestions for Authors

This paper is presented in reasonable 'english' given that the authors are not native english speakers.

The work follows from previous work (unpublished but on the arXiv)  and the first section is developed from that (analytical) approach, including equations and a diagram to be found there.  The idea is that energetic electrons and positrons, resulting from the annihilation of dark matter, Compton scatter galactic starlight from M31 which is observable from the Milky way as energetic gamma-rays.  However, the point of the paper is to note that the inclination of M31 may result in an anisotropic distribution of those gammas 'above' and 'below' the galaxy.  This original approach is extended later in the paper to make a more complete calculation of the gamma-ray anisotropy which results in a less obvious anisotropy.

The paper also follows from previous work, which would justify assumptions such as an electron energy of 1.8 TeV from dark matter annihilation.  Such assumptions would be better with a brief description of their origin within the paper.  This also applies elsewhere such that the authors could help the reader to better understand (within the paper) why particular values are chosen.  And resulting values found for various parameters - a little more 'physical' discussion would generally be helpful in the paper.

The paper is interesting and topical.  It is generally clearly written.  It concentrates on two photon/electron energies only (although Figure 2 etc. cover a broad energy range).  In this sense, the paper is selective but the physical arguments are usually clearly discussed when discussions are presented.

The assumed diameter of M31 appears to be rather high (80 kpc), and the dimensions of a dark matter halo (radius (?) 100 kpc) is not justified here.  The low energy photons are assumed to come from the whole galactic disk (line 181).  This will not be uniform and emission will surely be a function of distance from the galactic center.  (As an aside, observations of a possible anisotropy will also need to consider any contribution from M32 nearby with its supermassive black hole and its own starlight).  Considerations of the dark matter halo dimensions are important and should significantly affect the results of the paper.  Some discussion would be welcome here.

I didn't follow the values of some of the numbers in tables 1 and 2.  The text could clarify these.  "5.10^-1" in section 5 of row 1 of table 1 contrasts with "5.10^-11" in section 4.  In table 2, Rphi of 10^9 is appreciable larger than 45 in table 1.  The reader would appreciate more discussion of these figures and their importance.

As a small comment, the title used "DM".  For a title, the full statement "dark matter" would be more appropriate.  Also, though it is of practical use, lines 201-204 change from dimensions of M31 in pc to dimensions in degrees.  I would prefer to retain local dimensions (pc).  In fact, I did not really understand these four lines and this may be my only real problem with 'english' in the paper.

Comments on the Quality of English Language

The English Language in the paper is generally good enough to avoid misunderstanding.  As noted above, I did have a problem with lines 201-204.

Author Response

We are thankful to reviewer for their report.

We included all the minor corrections required, they are highlithed by green colour in the text.

We are gratefull again, these reviewer's remarks will only help to improve the quality of the manuscript.

With the best regards

Authors

Reviewer 2 Report

Comments and Suggestions for Authors

The nature of dark matter in galaxies is still unclear. The physicists propose various candidates into the dark matter media among hypothetical elementary particles but no experimental results concerning detecting any of these particles are obtained. Hence any propositions of new ways to detect signatures of dark matter particles reactions are valuables. The authors analyze a possibility to detect electron-positron production in dark matter decay through the mapping of the M31 galaxy halo in gamma-rays. The analysis has revealed the negligible effect so the dark matter cannot be detected in such a way. But to close an ineffective approach, it is good contribution into the problem solution.

Author Response

We are grateful to reviewer's positive report, and sure that this work will justify their high assessment.

With the best regards

Authors